# Can Disruption of Basal Ganglia-Thalamocortical Circuit in Wilson Disease Be Associated with Juvenile Myoclonic Epilepsy Phenotype?

**DOI:** 10.3390/brainsci12050553

**Published:** 2022-04-26

**Authors:** Jessica Rossi, Francesco Cavallieri, Giada Giovannini, Francesca Benuzzi, Daniela Ballotta, Anna Elisabetta Vaudano, Francesca Ferrara, Sara Contardi, Antonello Pietrangelo, Elena Corradini, Fausta Lui, Stefano Meletti

**Affiliations:** 1Clinical and Experimental Medicine PhD Program, University of Modena and Reggio Emilia, 41121 Modena, Italy; jessicarossi.mail@gmail.com (J.R.); giovannini.giada@gmail.com (G.G.); 2Neurology Unit, Neuromotor & Rehabilitation Department, Azienda USL-IRCCS of Reggio Emilia, 42123 Reggio Emilia, Italy; 3Unit of Neurology, OCB, Azienda Ospedaliero Universitaria di Modena, 41126 Modena, Italy; annavaudano@googlemail.com (A.E.V.); stefano.meletti@unimore.it (S.M.); 4Department of Biomedical, Metabolic and Neural Sciences, University of Modena and Reggio Emilia, 41125 Modena, Italy; francesca.benuzzi@unimore.it (F.B.); daniela.ballotta@unimore.it (D.B.); fausta.lui@unimore.it (F.L.); 5Internal Medicine Unit, Centre for Hemochromatosis and Heredometabolic Liver Diseases, Policlinico, Azienda Ospedaliero Universitaria di Modena, 41124 Modena, Italy; ferrara.francesca@aou.mo.it (F.F.); antonello.pietrangelo@unimore.it (A.P.); elena.corradini75@unimore.it (E.C.); 6IRCCS Istituto delle Scienze Neurologiche di Bologna, UOC Neurologia e Rete Stroke Metropolitana, Ospedale Maggiore, 40133 Bologna, Italy; scontardi@gmail.com; 7Department of Medical and Surgical Sciences, University of Modena and Reggio Emilia, 41125 Modena, Italy

**Keywords:** myoclonic epilepsy, Wilson disease (WD), globus pallidus internus (GPi), thalamocortical circuit, fMRI

## Abstract

In this paper, we describe the multimodal MRI findings in a patient with Wilson disease and a seizure disorder, characterized by an electroclinical picture resembling juvenile myoclonic epilepsy. The brain structural MRI showed a deposition of ferromagnetic materials in the basal ganglia, with marked hypointensities in T2-weighted images of globus pallidus internus bilaterally. A resting-state fMRI study revealed increased functional connectivity in the patient, compared to control subjects, in the following networks: (1) between the primary motor cortex and several cortical regions, including the secondary somatosensory cortex and (2) between the globus pallidus and the thalamo-frontal network. These findings suggest that globus pallidus alterations, due to metal accumulation, can lead to a reduction in the normal globus pallidus inhibitory tone on the thalamo-(motor)-cortical pathway. This, in turn, can result in hyperconnectivity in the motor cortex circuitry, leading to myoclonus and tonic-clonic seizures. We suppose that, in this patient, Wilson disease generated a ‘lesion model’ of myoclonic epilepsy.

## 1. Introduction

Wilson disease (WD) is an autosomal recessive inherited disorder of copper metabolism, due to a mutation of the ATP7B gene, resulting in the pathological accumulation of copper in many organs and tissues [1]. The most common neurological symptoms of WD are movement disorders, including tremor, dystonia, myoclonus, parkinsonism, and ataxia, as well as dysphagia, dysarthria, and drooling [1]. The most common sites of brain abnormalities are putamen, caudate, globus pallidus and thalamus, where the accumulation of copper and other ferromagnetic materials results in hyperintense areas in T2-weighted sequences [2]. MRI examination frequently shows hypointensities in T2-weighted and susceptibility weighted imaging (SWI) images, and this finding seems to be more strictly correlated with iron deposition, rather than copper concentration [3]. Seizures are reported in 6.2–8.3% of WD patients and can be generalized and focal aware/impaired awareness seizures [4]. Their occurrence may be due to metabolic encephalopathy, direct copper toxicity, pyridoxine deficiency during penicillamine therapy, and the progression of the disease itself [4]. EEG features in WD patients are not specific and include changes in background activity, focal slowing, and focal/generalized discharges [4]. White matter lesions on brain MRI are usually associated with epilepsy in WD with a possible bidirectional relationship, as there are reports demonstrating a role for seizures in contributing to the enlargement of white matter lesions, as well as neurological deterioration [5]. Moreover, seizures could also be caused by Wilsonian cortical damage [5].

Although myoclonic seizures have been previously described (in the form of progressive myoclonic epilepsy, generalized myoclonus, periodic myoclonus, and multifocal myoclonus), [6] to the best of our knowledge, there are no reports on clinical and electroencephalographic pictures of specific myoclonic epilepsies in WD patients.

We report the case of a patient with Wilson disease and a seizure disorder, characterized by an electroclinical picture resembling juvenile myoclonic epilepsy (JME), one of the generalized genetic epilepsies of the adolescent. JME is characterized by juvenile onset of generalized tonic-clonic seizures, myoclonic jerks without loss of consciousness, absences, and photosensitivity, with EEG showing normal background activity with generalized bursts of rapid spike-and-wave (4–6 Hz), and without focal abnormalities on magnetic resonance imaging (MRI) [7]. However, recent analyses of multimodal MRI found a crucial role in the disruption of the connection between basal ganglia and thalamocortical circuit in the genesis of tonic-clonic seizures [8]. Therefore, we aim to evaluate the hypothesis that the accumulation of ferromagnetic materials in the basal ganglia has a role in the pathogenesis of the patient seizure phenotype.

## 2. Case Presentation 

A 35-year-old right-handed woman came to our attention, after an incidental finding of hepatic cirrhosis with ascites at the time of cesarean section delivery in her first pregnancy. Based on the low ceruloplasmin levels (9 mg/dL; reference range 20–60 mg/dL), elevated 24 h urinary copper excretion (>100 µg; reference range <40 µg), increased hepatic copper concentration (1480 µg/g; reference range <50 µg/g dry weight), and detection of two heterozygous pathogenic variants in the ATP7B gene (p.Arg969Gln and p.Thr977Met), a diagnosis of Wilson disease was made. Her medical history was notable for a first tonic-clonic seizure at the age of 16, in the absence of any prodrome, following which she had developed a clinical picture characterized by photosensitivity, generalized tonic-clonic seizures, and myoclonic jerks without loss of consciousness, predominantly occurring after awakening. Multiple EEG studies showed generalized bursts of rapid spike-and-wave of 4–6 Hz (Appendix A), so a diagnosis of juvenile myoclonic epilepsy was assessed, and the patient started lamotrigine at a dose of 300 mg/day, with good seizure control. The medication had been gradually withdrawn before her first pregnancy, with no seizure exacerbation. At the time of our evaluation, the neurological examination showed a slight distal postural and intentional tremor to the upper limbs and a minimal bilateral ataxia to the finger-to-nose test. Treatment with penicillamine up to 900 mg/day was started. After 5 years of treatment, the patient presented a reappearance of seizures and myoclonic jerks, so lamotrigine therapy was reintroduced. We performed a standard MRI, which documented a marked involvement of the basal ganglia, with hypointensities in the globus pallidus internus bilaterally, in T2-weighted sequences (Appendix A). We wondered if the patient epilepsy phenotype and WD were a mere coincidence, or rather the seizure disorder of the patient was related to the metabolic alteration of WD and to basal ganglia alteration. To better understand the ictogenesis in our patient, we performed a multimodal MRI study using voxel-based morphometry (VBM) and fMRI resting state analysis to investigate the structural and functional cortico-subcortical connectivity of the patient. A group of 30 healthy women (mean age 35.4 + 10 years), with no history of neurological or psychiatric diseases, were recruited as controls. At the time of the multimodal MRI study, the patient was in treatment with penicillamine 900 mg and lamotrigine 300 mg.

Voxel based morphometry (VBM) and resting-state fMRI protocols are shown in the supplementary text section. All participants gave their written informed consent to participate in the study, which was approved by the local Ethics Committee.

## 3. Results

### 3.1. Voxel Based Morphometry Results

No significant difference was found in the cortical/subcortical grey matter volume between the patient and controls.

### 3.2. EEG-fMRI Results

No significant epileptiform activity was recorded during scanning, neither any sign of drowsiness or sleep physiological activity.

### 3.3. Functional Connectivity of the Primary Motor Cortex

When compared to the controls, the patient’s left M1 showed increased functional connectivity with the pre- and post-central gyri bilaterally, including the right secondary somatosensory cortex (Figure 1; Table 1; Appendix A). Increased connectivity was also found between the patient’s right M1 and ipsilateral supramarginal gyrus, pre- and post-central gyri (Figure 1; Table 1; Appendix A). The left M1 showed decreased connectivity with the left parahippocampal gyrus, cuneus, and lingual gyrus (Table 1; Appendix A).

### 3.4. Functional Connectivity of the Supplementary Motor Area

No region of increased or reduced connectivity in the patient versus the controls was detected for SMA.

### 3.5. Functional Connectivity of the Globus Pallidus

The patient’s left and right globus pallidus (GP) showed increased functional connectivity with bilateral pre- and post-central gyri (medial portion), bilateral thalamus, the caudate nucleus (mostly the head portion; Figure 2; Table 1; Appendix A), compared to the controls. Decreased functional connectivity of GP with the body of the caudate nucleus and cerebellum was observed. The right GP was also less connected with the bilateral occipital lobe and right parahippocampal gyrus (Figure 2; Table 1; Appendix A).

## 4. Discussion

The patient showed altered resting-state functional connectivity of the motor cortex, as well as of the basal ganglia circuit, and between the latter and the thalamocortical relay. Notably, an overall increase in basal ganglia-motor system connectivity was observed. These functional changes were not paralleled by the modifications of the cortical and subcortical volumes in the VBM analysis. We can suppose that the accumulation of ferromagnetic materials in the basal ganglia, especially in the globus pallidus internus, led to a disruption of the GP inhibitory tone on the thalamocortical circuit, contributing to the observed increase in MRI functional connectivity in the basal ganglia-motor cortex loop. 

Recent computational analyses with multimodal MRI pointed out the presence of structural and functional alterations in patients with JME, particularly in the thalamus and thalamo-frontal network [9]. Several VBM studies showed alterations in the frontal and thalamic gray matter volume [9], and resting state fMRI studies demonstrated functional connectivity alterations in the prefrontal cortex and thalamo-frontal circuits [10,11]. Notably, Vollmar et al. (2012) demonstrated an increase in the connectivity between the prefrontal cognitive cortex and motor cortex in JME patients [10]. Focusing on the thalamo–frontal circuit, O’Muircheartaigh et al. (2012) demonstrated altered connectivity between the thalamus and medial frontal regions, during resting-state fMRI [11]. A growing body of evidence suggests that the structural and functional changes in epilepsy can also affect the basal ganglia [8], which have a crucial role in modulating thalamocortical projections and cortical excitability. It has been recognized that a structural or functional alteration of the basal ganglia can be at the basis of generalized spike-wave discharges [9] and can lead also to absence seizures, as well as to the evolution from focal to bilateral tonic-clonic seizures [8]. Among the basal ganglia, a growing body of interest is placed in the globus pallidus internus, which showed a marked alteration in our patient. Indeed, Kim J. H. et al. (2018) pointed out how JME patients tend to exhibit significant volume reductions in the left pallidus and bilateral putamen and thalamus, when compared to the controls [12]. Moreover, basal ganglia circuits have been demonstrated to have a role in the secondary generalization of focal seizures. He X. et al. (2020), through a series of “disconnection” simulations, demonstrated that an alteration of connectivity between the basal ganglia and the thalamo-frontal network is at the basis of focal to bilateral tonic-clonic seizures [8], and that this alteration depends upon the direct basal ganglia pathway. Specifically, the disruption in the striatum-modulated tonic inhibition of the thalamus, by the globus pallidus internus, is responsible for the greater susceptibility to secondary seizure generalization [8]. 

Indeed, this case-study has several limitations. First, beyond being a single case, direct evidence of a link between the increased functional connectivity and spike and wave generation in the patient was lacking. Unfortunately, at the time of the functional MRI experiment, no interictal epileptic discharges were evident. However, the observed increased functional connectivity can represent an enduring and predisposing condition, even in the absence of active epileptic discharges/epilepsy. This has indeed been demonstrated in JME patients [10,11]. Furthermore, the observed findings could be the expression of JME itself, and not a consequence of Wilson disease. 

Nevertheless, we assumed that the increased connectivity within the motor system, and between the globus pallidus and thalamo-frontal network, could be related to the possible role of the basal ganglia damage, due to metal accumulation in the pathogenesis of the patient seizure phenotype. In this view, our case may represent a lesion model of myoclonic epilepsy and of epilepsy with bilateral tonic-clonic seizures. 

## Figures and Tables

**Figure 1 brainsci-12-00553-f001:**
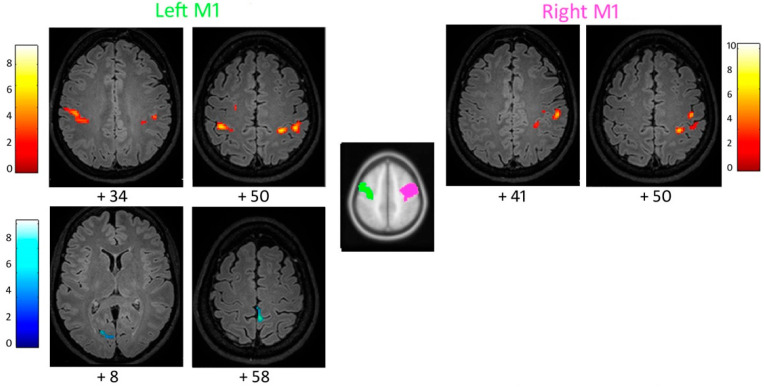
Increased (**top**) and reduced (**bottom**) functional connectivity of the patient’s left and right primary motor cortex M1, with respect to the control group (cluster size threshold k > 87 and k > 84, corrected at α < 0.05, respectively). Clusters are shown in the axial slices of the patient’s normalized flair 3D image.

**Figure 2 brainsci-12-00553-f002:**
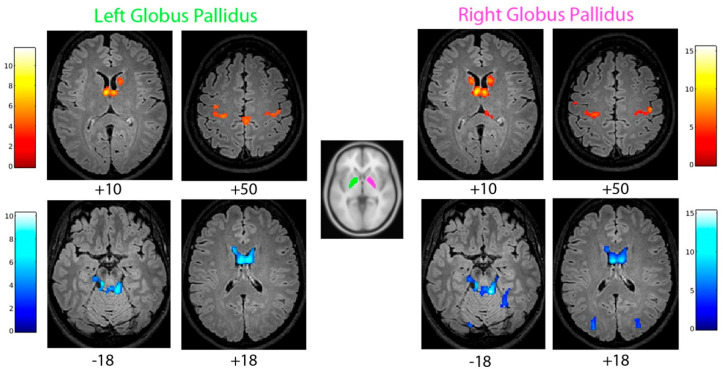
Increased (**top**) and reduced (**bottom**) functional connectivity of the patient’s left and right globus pallidus, with respect to the control group (cluster size threshold k > 95 and k > 108, corrected at α < 0.05, respectively). Clusters are shown in the axial slices of the patient’s normalized flair 3D image.

**Table 1 brainsci-12-00553-t001:** Increased and reduced functional connectivity of the patient’s regions of interest (ROIs), compared to the control group.

Regions of Interest (ROIs)	Increased Connectivity	Decreased Connectivity
Primary motor cortex (M1)	Left M1: with the pre- and post-central gyri bi-laterally, including right secondary somatosensory cortex. Right M1: with ipsilateral supramarginal gyrus and pre- and post-central gyri.	Left M1: with the left parahippocampal gyrus, cuneus and lingual gyrus.
Globus pallidus (GP)	Bilateral GP: with bilateral pre- and post-central gyri (medial portion), bilateral thalamus and the head of the caudate nucleus.	Bilateral GP: with the body of the caudate nucleus and cerebellum. Right GP: with bilateral occipital lobe and right parahippocampal gyrus.

## Data Availability

Not applicable.

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
