# Peer review of "Can Disruption of Basal Ganglia-Thalamocortical Circuit in Wilson Disease Be Associated with Juvenile Myoclonic Epilepsy Phenotype?"

_brainsci, 2022, doi:10.3390/brainsci12050553_

Round 1

Reviewer 1 Report

Title: Can Disruption of Basal Ganglia-Thalamocortical Circuit in Wilson Disease be Associated with Juvenile Myoclonic Epilepsy Phenotype?

The paper is describing very interesting case.

Patients with WD develop neurological symptoms which are not surprising. Epilepsy seizures are not common in WD. We know that penicillamine can lead to neurological deterioration and the progression of MRI changes. In this case, we have no illustration on anticopper therapy.

So, I suggest the authors explain more about anticopper therapy.

Also, I recommend that the authors cite and discuss this reference as well: "Kim, Y.E., Yun, J.Y., Yang, HJ. et al. Unusual epileptic deterioration and extensive white matter lesion during treatment in Wilson’s disease. BMC Neurol 13, 127 (2013)." https://doi.org/10.1186/1471-2377-13-127

I recommend publishing this paper after minor revision.

Reviewer 2 Report

Comments to Authors

Jessica Rossi, and colleagues have submitted a case report describing patient with Wilson Disease presenting as myoclonic epilepsy. The authors presented evidences to show possible link between Wilson disease induced copper deposition in the basal ganglia (Globus pallidus) with alteration of thalamo-frontal and thalam-ocortical network leading to pathogenesis of Juvenile Myoclonic Epilepsy.

Overall, this manuscript is well written, has an important clinical message, and should be of great interest to the readers.

Few suggestions: 

  1. Abstracts: No comments
  1. Introduction
  • The authors should consider mentioning few points from literature about myoclonic epilepsy seen in patients with Wilson Disease as described in six or more patients  (manuscript title: Coexistence of seizure with Wilson's disease: a systematic review and other relevant articles).
  1. Case: No comments
  1. Results – Although, it is very well written, the authors should consider making table to compare findings between patient and controls. 
  2. Discussion: No comments 
